1 Analysis of geostationary satellite derived cloud parameters associated with high ice water

| 2 | content | environments |  |
|---|---------|--------------|--|
| - | concent |              |  |

- 3
- 4 Adrianus de Laat<sup>1</sup>, Eric Defer<sup>2</sup>, Julien Delanoë<sup>3</sup>, Fabien Dezitter<sup>4</sup>, Amanda Gounou<sup>5</sup>, Alice
- 5 Grandin<sup>4</sup>, Anthony Guignard<sup>3</sup>, Jan Fokke Meirink<sup>1</sup>, Jean-Marc Moisselin<sup>5</sup>, and Frédéric Parol<sup>6</sup>

6

- <sup>7</sup> <sup>1</sup>KNMI, de Bilt, The Netherlands
- 8 <sup>2</sup>Laboratoire d'Aérology, CNRS/OMP Toulouse, France
- 9 <sup>3</sup>Laboratoire Atmosphère, Milieux et Observations Spatiales, UVSQ, Guyancourt, France
- <sup>4</sup>AIRBUS, Toulouse, France
- <sup>5</sup>Meteo France, Toulouse, France
- 12 <sup>6</sup>Laboratoire d'Optique Atmosphérique, Université de Lille, Sciences et Technologies,
- 13 Villeneuve d'Ascq, France
- 14
- 15 correspondence to: Adrianus de Laat (laatdej@knmi.nl)

#### 17 Abstract

18

We present a newly developed high ice water content mask (High IWC) based on measurements of the cloud physical properties (CPP) algorithm applied to the geostationary Meteosat Second Generation (MSG) Spinning Enhanced Visible and Infrared Imager (SEVIRI). The mask was developed within the European High Altitude Ice Crystals (HAIC) project for detection of upper atmospheric high IWC, which can be a hazard for aviation.

Evaluation of the High IWC mask with satellite measurements of active remote sensors of cloud 24 properties (CLOUDSAT/CALIPSO combined in the DARDAR product) shows that the High 25 IWC mask can be fine-tuned for detection of high IWC values  $> 1 \text{ g/m}^3$  in the DARDAR 26 profiles. The best CPP predictors of High IWC were the condensed water path, cloud optical 27 thickness, cloud phase, and cloud top height. The evaluation of the High IWC mask against 28 DARDAR provided some indications that the MSG-CPP High IWC mask is more sensitive to 29 cloud ice or cloud water in the upper part of the cloud, which is relevant for aviation purposes. 30 Biases in the CPP results were also identified, in particular a solar zenith angle (SZA) 31 32 dependence that reduces the performance of the High IWC mask for SZAs  $> 60^{\circ}$ . Verification statistics show that for the detection of High IWC a trade-off has to be made between better 33 detection of High IWC scenes and more false detections, *i.e.* scenes identified by the High IWC 34 mask that do not contain IWC > 1 g/m<sup>3</sup>. However, the large majority of these detections still 35 contain IWC values between 0.1-1 g/m<sup>3</sup>. 36

Comparison of the High IEC mask against results from the Rapid Developing Thunderstorm (RDT) algorithm applied to the same geostationary SEVIRI data showed that there are similarities and differences with the High IWC mask: the RDT algorithm is very capable of

- detection young/new convective cells and areas, whereas the High IWC mask appears to be
- better capable of detecting more mature and ageing convection as well as cirrus remnants.
- The lack of detailed understanding what causes aviation hazards related to High IWC hampers
- further tuning of the High IWC mask. Additional evaluation of the High IWC mask against field
- campaign data should provide more information on the performance of the MSG-CPP High IWC
- mask and contribute to a better characterization.

## 47 **1. Introduction**

Weather hazards can have a significant impact on aviation via disturbing flight schedules and 49 causing air traffic delays, but also as a cause of accidents, some of them fatal. Among the 50 51 weather related effects on aviation are changes in visibility like fog, clouds, rain, snow, hail, wind, turbulence, lightning, smoke and volcanic ash [Perkins et al., 1998; Bragg et al., 2002; 52 Mecikalski et al., 2007]. One particular hazardous process is in-flight or in-service icing. Aircraft 53 may penetrate clouds of super-cooled water droplets, or high densities of ice particles. The 54 droplets or particles may deposit on cold aircraft surfaces, affecting aerodynamic properties of 55 the plane, the engine performance, or inlets and nozzles used for onboard monitoring of 56 environmental conditions, the latter resulting in a malfunctioning of for example speed sensors 57 [Mason et al., 2006; Mecikalski et al., 2007]. 58

The microphysical conditions under which in-service icing may occur are generally well understood: either large amounts of super-cooled cloud droplets are present, or high 60 61 concentrations of ice particles. However, what the corresponding general atmospheric conditions are, and how to identify or diagnose them, has been a much more difficult task. This is in part 62 related to the notion that the majority of in-service icing events appear to occur outside of what is 63 64 called "classic" convection, *i.e.* areas of vigorous updrafts [Grzych and Mason, 2010]. Approximately only 20% of in-service icing events are associated with this "classic" convection. 65 The remaining 80% appears to be related to occurrence of ice crystals in anvils, with only weak 66 67 or moderate convection and turbulence. Such systems are characterized by large concentrations of small particles rather than large particles (hail) and/or super-cooled water droplets. 68

Furthermore, areas of large concentrations of small ice particles are difficult to detect by onboard radar, in contrast to the "classical" convection where the deep convection and vigorous cores can be detected by on-board radar and thus can be avoided. Finally, it appears that the majority of in-service icing events are occurring in tropical and subtropical regions of the world (30N-30S), although potential in-service conditions can occur at higher latitudes [Grzych and Mason, 2010].

To address many of the issues related to in-service icing events and in anticipation of regulation changes regarding mixed phase and glaciated icing conditions, the large European High Altitude Ice Crystals project (HAIC) was initiated in 2012 to investigate a wide range of aspects of inservice icing. The HAIC project combines laboratory experiments (wind chambers), field campaigns, numerical modeling and remote sensing techniques to study a variety of aspects of in-service icing.

Laboratory measurements focus on the characterization, optimization, enhancement and 81 82 selection of the most sophisticated cloud microphysics probes in order to measure mixed phase and glaciated icing conditions during flight tests and to calibrate icing wind tunnels. 83 84 Furthermore, the HAIC project aims at measurement and characterization of the microphysical properties of core or near-core regions of deep convective clouds, including cloud liquid and ice 85 water contents and particle size and shape distributions. Finally, HAIC aims at characterizing the 86 87 atmospheric conditions for possible in-service icing and the detection of such areas in satellite remote sensing products. 88

An important aspect of the HAIC project is the development of space-borne remote detection and
 now-casting application of glaciated icing conditions based on imagery of geostationary MSG SEVIRI (Meteosat Second Generation - Spinning Enhanced Visible and InfraRed Imager)

satellite observations. Atmospheric conditions under which in-service icing occurs are not well
understood nor well characterized – which is also the prime justification for the HAIC project.
However, it is widely accepted that the presence of High Ice Water Content is a crucial condition
for the occurrence of in-service icing. Detection of areas of potential High Ice Water Content by
satellite remote sensing thus provides important spatio-temporal information for the possible
occurrence of in-service icing events.

In this paper, we present a new High Ice Water Content mask (High IWC mask) based on the 98 output of the MSG - Cloud Physical Properties (CPP) algorithm [Roebeling et al., 2006; Meirink, 99 2013] that was developed within the Climate Monitoring Satellite Application Facility (CMSAF) 100 101 of the EUropean METeorological SATellite organization (EUMETSAT), and applied to the geostationary SEVIRI satellite measurements. The CPP algorithm provides a number of cloud 102 physical properties that are of interest in diagnosing possible in-service icing conditions. The 103 104 high IWC mask will be derived and evaluated against measurements of cloud properties from active remote sensing instruments on board of satellites, and finally the high IWC mask will be 105 compared with the EUMETSAT SEVIRI Rapid Development Thunderstorm (RDT) product that 106 107 is used to identify rapidly growing thunderstorms and convective systems [Autonès, 2012].

2. Project description and datasets used.

**2.1 HAIC**

Within the European FP7 HAIC project, academics and aeronautic industries are collaborating
within six main research activities that include dedicated field campaigns, development of new in
situ probes, space-based detection and monitoring, upgrade of on-board weather radars,

improvement of ground test facilities, and modeling of melting and impingement processes. All

activities are designed to enhance aircraft safety when flying in mixed phase and glaciated icing

conditions.

The HAIC Sub-Project 3 (SP3), entitled Space-borne Observation and Nowcasting of High Ice 118 Water Content Regions, focuses on the development of space-borne remote detection of high Ice 119 120 Water Content (IWC) and nowcasting techniques to support the second and third HAIC flight campaigns and ultimately provide relevant near real-time weather information through Air 121 Traffic Management (ATM). The SP3 investigations are divided in three interacting Work 122 123 Packages (WPs): (i) Geostationary space-borne retrievals of high IWC events focusing on the 124 detection of high IWC cloud regions mainly from the SEVIRI imager on MSG in daytime; (ii) Polar orbiting space-borne retrievals of high IWC events investigating the detection of high IWC 125 cloud regions from visible, infrared and microwave passive and active observations of the space-126 based A-Train mission; (iii) Nowcasting of tropical convection dedicated to the tracking of deep 127 128 convection over the Tropical Atlantic for operational applications based on the Rapid Development Thunderstorm (RDT) nowcasting tool. Following the HAIC Technology Readiness 129 130 Level (TRL) strategy, the SP3 activities are required to pass with success three TRL levels: TRL3 (characteristic proof of concept), TRL5 (breadboard validation in relevant environment) 131 132 and TRL6 (prototype demonstration in a relevant environment).

2.2 MSG-CPP

The CPP algorithm [Roebeling et al., 2006; Meirink, 2013] uses SEVIRI's visible (VIS) and near-infrared (NIR) measurements to retrieve cloud optical thickness ( $\tau$ ) and cloud particle

effective radius (r<sub>e</sub>) by applying the classical Nakajima and King [1990] approach. This approach 139 is based on the basic feature that the reflectance at a for cloud particles non-absorbing wavelength is primarily related to  $\tau$ , while the reflectance at an absorbing wavelength is mainly 140 related to re. For SEVIRI retrievals the VIS 0.64 µm and the NIR 1.63 µm channels have been 141 used here as non-absorbing and absorbing channels, respectively. Around 1.63 µm ice particles 142 143 are more absorbing than water droplets, which is not the case at 0.64 µm. Hence, together with the use of a thermal infrared (IR) window channel to inform on cloud top temperature, this 144 145 allows to retrieve cloud thermodynamic phase.

CPP is based on look-up tables (LUTs) of top-of-atmosphere (TOA) reflectances for singlelayer, plane parallel, water and ice clouds, simulated by the Doubling Adding KNMI (DAK) 147 radiative transfer model [Stammes, 2001]. Single scattering properties have been calculated 148 using Mie theory for spherical water droplets and ray tracing for imperfect hexagonal ice crystals 149 [Hess et al., 1998], respectively. Absorption by atmospheric trace gases is taken into account 150 based on Moderate Resolution Atmospheric Transmission code simulations (MODTRAN4 151 Version 2 [Anderson et al., 2001]). For cloudy pixels (cloud contaminated or cloud filled, as 152 153 determined by the cloud mask described in Roebeling et al. [2006])  $\tau$  and r<sub>e</sub> are retrieved by 154 matching the observed reflectance to the LUTs. First the ice cloud LUT is tried. If this leads to a 155 match and if the cloud top temperature – retrieved from the 10.8 µm channel - is below 265 K, 156 the thermodynamic phase is set to ice. Otherwise, the water cloud LUT is used, and the phase is set to liquid. Liquid and ice water path (LWP and IWP) are then calculated following Stephens 157 [1978]: 158

$$LWP = \frac{4}{3Q_e}\rho_l r_e \tau; \qquad IWP = \frac{4}{3Q_e}\rho_i r_e \tau \qquad (1)$$

# 162 where $Q_e$ is the extinction efficiency at visible wavelengths (set to 2), and $\rho_l$ and $\rho_i$ are the densities of water and ice, respectively. Eq. (1) assumes a vertically homogeneous distribution of 163 cloud condensate. CPP uses surface albedo at the VIS and NIR channels based on MODIS 164 [Moody et al., 2005], and water vapour path from the ERA-Interim reanalysis project [Dee et al., 165 2011] of the European Center for Medium range Weather Forecast (ECMWF) as ancillary input 166 167 data. Cloud property retrievals become very uncertain at high solar zenith angles ( $\theta_0$ ) and viewing zenith angles ( $\theta$ ). Therefore, no retrievals are performed for $\theta_0 > 78^\circ$ or $\theta > 78^\circ$ . Earlier 168 169 versions of CPP have been extensively validated using ground-based observations [Roebeling et 170 al., 2008; Wolters et al., 2008) and used for the evaluation of regional climate models [Roebeling and van Meijgaard, 2009; Greuell et al., 2011]. Note that the CPP parameters associated with 171 reflected solar radiation are mostly representative for the upper parts of clouds, in particular in 172 case of optically thick clouds (e.g. deep convection). Because of the reliance of CPP on reflected 173 174 solar radiation most of the photons are reflected back from the upper parts of optically thick clouds [Platnick, 2000]. Hence, little information from deep within optically thick clouds can be 175 176 obtained. Also note that unpublished results indicate that CPP Reff does not correlate very well with remote sensing profiles of Reff. Furthermore, the physical interpretation of Reff is rather 177 178 complicated and care should be taken with interpreting Reff as representative for real-world cloud 179 particles sizes, in particular for ice clouds [e.g. McFarquhar and Heymsfield, 1998; Mitchell et al. 2011]. Detailed information on the CPP version used can be found in Meirink [2013]. 180 181

**2.3 DARDAR**

DARDAR (raDAR/liDAR) [Delanoë and Hogan, 2008, 2010] consists in two synergistic products derived from the combination of the CloudSat radar [Stephens et al., 2002] and 185 CALIPSO lidar [Winker et al., 2009] measurements. These products are distributed through the 186 ICARE centre in Lille (France). The first one, DARDAR-MASK [Delanoë and Hogan, 2010, 187 Ceccaldi et al 2013], is mainly a target classification of the scene observed by both CloudSat and 188 CALIPSO. More precisely the DARDAR-MASK data set employs a combination of the 189 CloudSat, CALIPSO measurements to identify cloud, precipitation and aerosol presences and 190 also retrieve cloud phase properties. The algorithm, based on a decision tree, was originally 191 designed to identify ice clouds on the basis of the synergy of surface-based radar, lidar 192 193 observations. The DARDAR-MASK returns a range of categories: clear, ground, stratospheric features, insects, aerosols, rain, super-cooled liquid water, liquid warm, mixed-phase and ice. 194 The algorithm also permits an "unknown" classification when it is not possible to determine one 195 of these categories [Delanoë and Hogan, 2010]. This commonly occurs in regions where the 196 197 radar and lidar signal have been heavily attenuated or are missing. DARDAR-MASK used CALIPSO backscatter and temperature to identify super-cooled water in the 0°C to -40°C range 198 199 [Ceccaldi et al 2013], while the depolarization is considered too noisy to be used at the CALIPSO resolution [Delanoë and Hogan, 2010]. 200

Ice cloud properties are available in the second DARDAR-CLOUD product [Delanoë and Hogan, 2010]. This product uses the "varcloud variational technique" [Delanoë and Hogan, 2008] which combines the CloudSat radar and CALIPSO lidar profiles for retrieving the extinction coefficient, IWC and Re of the ice cloud. DARDAR-CLOUD assumes a "unified" PSD given by Delanoë et al. [2005, 2014]. The mass-size and area-size relations of non-spherical