# Peer review of "Analysis of geostationary satellite derived cloud parameters associated with high ice water"

_Atmospheric Measurement Techniques, 2016_

## Referee Comment (RC1) · Anonymous Referee #1 · 27 Nov 2016

This paper represents an important contribution to our understanding of the capabilities to detect high ice water conditions that present a hazard to aviation operations. The authors demonstrate the use of SEVIRI satellite products combined to identify HIWC areas, and quantify the performance of their HIWC Mask product. Their methods are scientifically sound, and the paper explains their methods and results in a clear and logical way.

General comments are as follows: 1. What is the purpose of the MSG-CPP comparison with DARDAR described in Section 3.2? It is explained later in the paper that thresholds for the HIWC Mask product are optimized using DARDAR paper, but it isn't clear how the results of the comparison are used for this purpose. 2. The impact of the

height of the DARDAR high IWC maximum on the HIWC Mask is mentioned on p.19 (line 389-391) and again on p. 21 (line 436-439). These two statements appear to be somewhat contradictory. Can you clarify? 3. Regarding the HIWC Mask comparison with RDT, it is concluded that the larger the cell, the larger the fraction of pixels identified. On how many cases is this conclusion based? 4. It would be interesting to include some discussion of how the HIWC Mask product might be used in an operational environment. Could you comment on the types of users who might benefit, the timeliness of the product versus the time scale of the physical processes, and the impact of the current level of uncertainty, for example?

Specific comments: 1. It's rather difficult to distinguish the blue outlines and orange dots in Figure 10. 2. Line 40, replace "detection" with "detecting" 3. Line 42, replace "lack of detailed understanding what causes..." with "lack of detailed understanding of what causes..."
* * *

---

## Referee Comment (RC2) · Anonymous Referee #2 · 4 Dec 2016

Formal Review Of: Analysis of geostationary satellite derived cloud parameters associated with high ice water content environments

Lead Author: A. de Laat

This paper describes methods for discriminating high ice water content (HIWC) producing convection using satellite imager observations and satellite cloud microphysical retrieval products. The methods are threshold-based with the first set of thresholds being derived from in-service aircraft encounters of ice water content (IWC) > 1 g/m3 and the second being an adjustment of these thresholds to maximize Critical Success Index based on IWC derived from CloudSat and CALIPSO observations (their so-called DARDAR product). Satellite parameters used for HIWC discrimination include cloud

mask, effective radius, cloud phase, cloud temperature/height, cloud optical thickness, and ice water path.

As noted in my initial review, I found this paper to be well written and it is clear that the set of authors have extensive knowledge of the HIWC engine icing phenomenon and have a clear understanding of the challenges associated with HIWC detection/nowcasting. The authors make it clear that no passive satellite imager based algorithm will be capable of 90+% POD and 10% FAR (for example) due to the complex physics of the problem and that passive sensors cannot "see into the cloud" to altitudes where HIWC is located. Thus we are left to rely on inferences of in-cloud dynamics from cloud top observations. Nevertheless, even after their minor revisions to the paper and responses to my initial comments, I still do not feel that this work is ready for publication.

I have several significant concerns regarding this study that were not satisfactorily addressed from my initial review: 1) Your Equation 1 shows that ice water path is a parameterization based on optical depth and effective radius but if effective radius is not useful, then you are basically including optical depth twice. So your "optimized" end product includes almost any pixel that could possibly be ice (cloud temp < 270), optical depth > 20, and ice water path > 100. The water path would almost never be less than 100 with optical depth of > 20, even with very small ice radii. Please explain your rationale for essentially double use of optical depth. I could envision other parameters that would be of value to substitute for one of your optical depth based parameters.

I appreciate the challenges you state regarding isolating HIWC in clouds and lack of understanding of how HIWC is generated. But this mask does not seem to be a product that the aviation community could use. No forecaster I know would alter flight routes based on this type of product The product seems more like a "ice cloud mask for moderate to high optical thickness cloud" than anything that could be useful for discriminating localized HIWC conditions. You're basically showing locations where HIWC would not be rather than where exactly it is. I have analyzed how your product would perform

using every flight from the HAIC-HIWC project. I find that with your mask thresholds and use of TWC > 1 to define HIWC, your product would provide a 0.97 POD and 0.78 FAR In addition, if you're limited to solar zenith angle of < 60 then, you'd only have a product for a handful of hours a day, and only a few of these hours would actually have robust convection (i.e. intense storm activity typically begins no earlier than 2 PM). Given what I explain here and the thresholds you've selected as "optimal", please help me to understand how this product could be useful to the operational forecast community. Would you honestly expect aviators to fly around all the moderate optical thickness clouds you identify to "avoid HIWC"?

2) I still do not find the reasoning behind including RDT to be well explained nor are the findings worthwhile. Please explain the POD and FAR for RDT identification of HIWC. We know that based upon your optical depth > 20 and cloud temp < 270 optimized HIWC mask thresholds, almost every convective anvil will be identified. Based upon the name of the product, RDT is designed to identify rapidly growing thunderstorms. This rapid development often occurs during the initial development phase where, as you indicate, the cells are small. So of course you should see some mismatch in the areal coverage of the product. You state that you are "comparing" the two products but the one scene you show isn't especially interesting aside from showing the areal coverage differences. In examples I personally have seen in talks and papers, RDT seems quite capable of detecting large anvils and embedded developing regions within them, not just newly developing small cells. I would think that, given how this product behaves and that it can extract cloud cooling and areal expansion, RDT may actually be better suited for HIWC detection than the mask you show here. Perhaps merging the two products (i.e. cloud temp < 270, optical depth < 20, within a cooling/expanding RDT object) would create a product more accurate and useful to forecasters.

In summary, given the relatively weak analysis shown in the RDT section and tangential relationship to the primary focus of the paper (as it is currently written), I am not seeing how this section is relevant and request again that the RDT section be removed to

improve the focus of the paper.

Given what I describe above and below, I do not feel that what seems to be the core goal of the paper, namely the development, validation, and characterization of an HIWC mask was satisfactorily accomplished here. The authors raise many interesting points and are clearly excellent scientists. Yet the extraneous material throughout such as RDT and DARDAR-CPP comparisons, combined with double usage of optical depth in the mask and an end "optimized" mask product that basically classifies any moderately thick ice cloud as HIWC are major roadblocks for me that I don't think could be addressed in the limited time allotted for revision. Nevertheless, I will allow the authors one final opportunity to address the issues I raise. As I am already on the fence between major revision and reject, if the core issues I mention throughout are not addressed, I will reject the next round.

Additional comments Line 37, replace IEC with IWC

Line 58, recommend citing Smith et al. (JAMC, 2012) here

Lines 65-66, you note that 80% of in service events were related to ice crystals in anvils and 20% related to strong convection. Do you think that this is caused by the fact that aircraft are directed to avoid classic convection by air traffic control and via signals they see on their onboard radar. I'm guessing that if there were more convective core penetrations, your perception of where in service events are located would change. Please comment on this in the paper text.

Line 80, please reference the partnership with North Americans via the HAIC-HIWC campaigns.

Line 85, if it is understood that 80% of events are outside of classical convection, then why was it a focus for HAIC to study core and near core regions?

Line 93, it seems as though Grzych and Mason (2010) and Grzych et al and Brevin et al (SAE, 2015) have characterized in service icing pretty well. This contradicts your

statement here that indicates that icing is not well understood or characterized. I recommend you revise your wording here.

Line 120, what are the "second and third HAIC flight campaigns?" I wouldn't expect many to know about this so I recommend you explain further or omit

Line 124, why is daytime only the focus here? Many flights occur at night especially over oceanic waters. Convection peaks in intensity at night over ocean and also over land associated with mesoscale convective systems that develop overnight. I understand that you want to utilize optical information to aid detection, and night would have IR only. Given what you say about the 60 degree solar zenith angle limitation, you're quite limited in the hours of day when your product can operate and intense convection is present. I request that you explain these additional limitations and prospects for night-time HIWC identification.

Lines 176-180, so are you saying here that the phenomenon of "small ice crystals" in storms commonly attributed to HIWC cannot be retrieved using satellite measurements? I see inclusion of Reff in your Mask Version 1, what is the purpose of including this field? I also recommend that these sentences be separated into another paragraph given that it is in an already long paragraph.

Line 184, please explain how these products differ from the "official" merged CALIPSO/CloudSat cloud microphysical products produced by the CloudSat Science Team (http://www.cloudsat.cira.colostate.edu/data-products/level-2c/2c-ice) and why you opted to use DARDAR rather than these? What are the advantages of DARDAR?

With regards to DARDAR, I read in Austin et al (JGR, 2009) that the uncertainty of the CloudSat IWC retrieval can be up to 40%. Given this uncertainty, what you perceive to be an HIWC event may not be, and vice versa. This casts doubts on the data used as foundation for HIWC truth. I strongly request you address this in the paper text.

Also with your DARDAR discussions, you're mentioning height of maximum IWC quite a bit, i.e. line 438, Figure 9. If you look at Figures 1-5 of Setvak et al. (Atmos Res, 2013), you'll see that in and near deep convective core regions, the CloudSat signal attenuates in the layer nearest cloud top (their Fig 5 is the best example) which is thought to be due to the presence of graupel near cloud top in strong updrafts. So the apparent height of max IWC would be very high, but in actuality, this height may be further down into the cloud that cannot be seen by CloudSat. I strongly recommend you consider this in how you state your results and mention this observational characteristic in the paper text.

Line 244, please provide a point of contact or a mechanism for one to acquire a list of these AIRBUS events for scientific research. I see an AIRBUS representative on the author list so I'd imagine that he could provide this information. The results from published journal articles are supposed to be reproducible and, without this list, one cannot verify the reproducibility of your results

Line 255, you spend some time in the Supplemental section doing a pretty good job of presenting an interesting HIWC mask (I'd say a more interesting one than v2 mask), but then in lines 254-257, you almost immediately cast off both the training dataset and mask as being not useful. This seems very odd. . .why bother with describing the mask if you don't think the in service events are useful?

Line 258, "was agreed upon", by whom? Please reword for clarity

Line 266, you say that a test dataset from 2008 was provided, but then only 31 orbits were analyzed? This is confusing; many more than 31 orbits were recorded in the entire year of 2008. Please explain in the paper text

Line 270, Figure 1, you use a combination of red and green to show where ice was observed in the profile. Red and green are an awful color combination for those that are color blind so I request that you consider this and recreate this and other figures that use red/green combos to highlight meaningful results.

Lines 270-328, I find that the material presented in these lines is extraneous and does not necessarily advance your goal of developing a HIWC mask. 278-288 talks about cloud top height assignment relative to DARDAR. Cloud top height was irrelevant to your optimized mask. Also, you spend a good bit of text comparing CPP to DARDAR IWC/IWP and effective radius considerations. Your goal in this paper was to develop a mask to discriminate HIWC using satellite observations. It really doesn't matter what CPP produces for CWP; all that matters is that you pick a threshold that gets the job of discriminating done well.

As I'm reading the text prior to line 270, I'm engaged with the paper and its message, then line 270 comes along and I lose focus, waiting for the main point that never seems to come. Please explain the relevance for including this material and perhaps make more concise to stay on the primary message of the paper.

Lines 348-353, I recommend rewording this paragraph as I'm not currently following what you're getting at here. You say "were then analyzed", what are you referring to? I don't see anything about steepness in the paper. Line 351 says "mask in the maximum IWC interval in the interval..." which reads a bit awkwardly.

Lines 376-382, Figure 8a, I see that you have a ROC curve here, but your axes are backwards. FAR is usually along the Y-Axis so that the "area under the curve" can be analyzed to assess the quality of a product. I recommend you reverse the axes here to make this plot consistent with how others in the validation community present their results.

Line 401, as noted above, if one uses your thresholds to analyze product performance relative to 45 second averaged TWC > 1 from the IKP2 probe during HAIC-HIWC Darwin, you get a product with an estimated POD of 0.97 and FAR of 0.78. Obviously we're not using CPP results here but other cloud retrieval algorithms have been proven to provide comparable retrievals to CPP as evidenced in the Hamann GEWEX assessment paper. This is quite inconsistent with the results you get here which is a really

big issue to contend with. How do you reconcile these differences between DARDAR based truth and IKP based truth?

Line 443-448. You mention that maximum IWC of 0.1 is "still quite high". My assessment of in-situ TWC data during HAIC-HIWC and column max IWC from RASTA retrievals is that 0.1 is about the 25th percentile of TWC observed during the HAIC/HIWC campaigns. My personal opinion is that 25th percentile is not particularly high.

Line 452, you may be seeing attenuation of CloudSat signal leading to higher IWC altitudes, which would indicate a more vigorous storm that could be easier to detect via satellite.

Line 470, I recommend a different term than "air masses" here

Line 563, I wouldn't exactly characterize the analysis here as "fine tuning" given the very liberal thresholds and double counting of optical depth used within the end product

Line 577, after thunderstorms have ceased developing and they are decaying, wouldn't the risk of HIWC greatly decrease? So the fact that the HIWC mask identifies these may not be a good thing, correct?

Line 603, where was the HAIC field campaign in 2016?

---

## Author Comment (AC1) · 31 Jan 2017

This paper represents an important contribution to our understanding of the capabilities to detect high ice water conditions that present a hazard to aviation operations. The authors demonstrate the use of SEVIRI satellite products combined to identify HIWC areas, and quantify the performance of their HIWC Mask product. Their methods are scientifically sound, and the paper explains their methods and results in a clear and logical way.

General comments are as follows:

1. What is the purpose of the MSG-CPP comparison with DARDAR described in Section 3.2? It is explained later in the paper that thresholds for the HIWC Mask product are optimized using DARDAR paper, but it isn't clear how the results of the comparison are used for this purpose.

*It is important to realize that this is the first time that a dedicated satellite-based High IWC product is presented. High Altitude Ice Crystals as a research topic is fairly new, and only emerged as a possible aviation hazard in the late 1990s and early 2000s.*

*Furthermore, research, results and understanding have grown organically within the HAIC project and the dedicated work packages on satellite measurements and detection of High IWC environments.*

*The SEVIRI CPP High IWC mask thus is a totally new product, and readers may not have general knowledge about for example DARDAR. We felt that introducing DARDAR without much more than a reference to other papers would be rather meager for those readers, so to speak.*

*Similarly, only during the last couple of years some intercomparison studies have appeared that compared measurements of cloud properties from passive (like SEVIRI) and active sensors (like DARDAR). Much of the work on identifying agreement and differences between those measurements is still work in progress, and often performed within the context of specific applications.*

*Because of these reasons, we thought it useful to spend some time on introducing DARDAR and presenting some general results about how DARDAR compares to SEVIRI-CPP. Like we similarly felt that it would be useful to also present the results on how the CPP High IWC mask v1 was constructed, which can be found in the SI.*

*[SUGGESTION]: we would be willing to consider putting section 3.2 in the SI, rather than in the main body, and only briefly discuss results, like now is done for the SEVIRI-CPP High IWC mask v1. With the notion that we prefer to keep section 3.2 in the main text.*

2. The impact of the height of the Cloud high IWC maximum on the HIWC Mask is mentioned on p.19 (line 389-391) and again on p. 21 (line 436-439). These two statements appear to be somewhat contradictory. Can you clarify?

*The discussion on p.19 (table 3) refers to CPP parameter threshold combinations with a CSI > 0.35, the discussion on p.21 to the performance of the mask itself. Reason for table 3 was to check if the highest CSI values somehow might show some (in) consistency among the parameter threshold values.*

*We removed table 3, as it is difficult to interpret, and simply note that different CPP parameter threshold combinations yield similar CSI performance statistics, and that we chose a set of CPP parameter thresholds that are most inclusive (= at the 'conservative' end of each CPP parameter threshold).*

*That way, the only location where the height is discussed is in the sensitivity of the performance of the High IWC mask to the actual height of the maximum IWC (p21).*

3. Regarding the HIWC Mask comparison with RDT, it is concluded that the larger the cell, the larger the fraction of pixels identified. On how many cases is this conclusion based?

*There turn out to be 12,416 such cases.*

*Number added to caption of figure 11.*

4. It would be interesting to include some discussion of how the HIWC Mask product might be used in an operational environment. Could you comment on the types of users who might benefit, the timeliness of the product versus the time scale of the physical processes, and the impact of the current level of uncertainty, for example?

*Although understandably it is an interesting point to discuss how the mask could be used (we are talking about a potentially hazardous environment for aviation), we are reluctant to discuss such application WITHIN the paper, in part because there appears to be disagreement about its operational usefulness.*

*Nevertheless, we can address some issues here:*

- *Currently the mask is provided in near-real-time by KNMI for the SEVIRI (Meteosat-10) domain (Europa, Africa).*
- *The mask has been successfully tested operationally for HIMAWARI (East Asia, Australia) as part of support for a HAIC 2016 field campaign in Australia, but it is currently not online as there are no immediate users.*
- *The mask can be easily applied to SEVIRI on Meteosat-8, which has been relocated to the Indian Ocean. We expect to be able to soon test CPP and thus the mask on GOES-R (Americas) in 2017.*

*In terms of operational use:*

- *Latency of SEVIRI (Meteosat10) mask is 15-30 minutes*
- *Latency could be shortened by parallel processing.*
- *Latency could be improved by applying motion vectors to make short-term forecasts (test will be performed in 2017). Reliability of motion vectors up to 60 minutes ahead of time is sufficient.*
- *Strategical use of mask by pre-plan flights around potentially hazardous areas*
- *Tactical use of mask by in-flight modifications of flight plan according to latest developments*

*For the last two applications we are part of a project proposal (2017) for a field test of in-flight guidance and route dynamic planning system, which should help to gain experience in providing CPP data (incl. the High IWC mask) to pilots.*

*We also started discussions with the Dutch KLM airliner about improved or new services for their daily operations, which the mask could be part of.*

*In the end, like with convection, the aviation authorities have to decide on what is acceptable and what is not. Current aviation regulations for thunderstorms show that there are no absolute restrictions – although there are clear recommendations how to handle thunderstorms. Airliners have considerable room to decide for themselves, and their decisions depend on safety, cost, and comfort. A High IWC mask – like many other environmental parameters - would be just one of the many parameters for them to consider. But it could easily become a part of a dynamical time-dependent 3D flight planning/guidance tool.*

*Future discussions, projects, and tests, with commercial aviation partners will show its true value.*

Specific comments:

1. It's rather difficult to distinguish the blue outlines and orange dots in Figure 10.

*Removed the orange dots, increase the thickness of blue contours.*

2. Line 40, replace "detection" with "detecting"

*Changed*

3. Line 42, replace "lack of detailed understanding what causes…" with "lack of detailed understanding of what causes…"

*Changed*

---

## Author Comment (AC2) · 31 Jan 2017

Formal Review Of: Analysis of geostationary satellite derived cloud parameters associated
with high ice water content environments

Lead Author: A. de Laat

This paper describes methods for discriminating high ice water content (HIWC) producing convection using satellite imager observations and satellite cloud microphysical retrieval products. The methods are threshold-based with the first set of thresholds being derived from in-service aircraft encounters of ice water content (IWC) > 1 g/m3 and the second being an adjustment of these thresholds to maximize Critical Success Index based on IWC derived from CloudSat and CALIPSO observations (their so-called DARDAR product). Satellite parameters used for HIWC discrimination include cloud mask, effective radius, cloud phase, cloud temperature/height, cloud optical thickness, and ice water path.

As noted in my initial review, I found this paper to be well written and it is clear that the set of authors have extensive knowledge of the HIWC engine icing phenomenon and have a clear understanding of the challenges associated with HIWCm detection/nowcasting. The authors make it clear that no passive satellite imager based
algorithm will be capable of 90+% POD and 10% FAR (for example) due to the complex physics of the problem and that passive sensors cannot "see into the cloud" to altitudes where HIWC is located. Thus we are left to rely on inferences of in-cloud dynamics from cloud top observations. Nevertheless, even after their minor revisions to the paper and responses to my initial comments, I still do not feel that this work is ready for publication.

I have several significant concerns regarding this study that were not satisfactorily addressed from my initial review:

1) Your Equation 1 shows that ice water path is a parameterization based on optical depth and effective radius but if effective radius is not useful, then you are basically including optical depth twice. So your "optimized" end product includes almost any pixel that could possibly be ice (cloud temp < 270), optical depth > 20, and ice water path > 100. The water path would almost never be less than 100 with optical depth of > 20, even with very small ice radii. Please explain your rationale for essentially double use of optical depth. I could envision other parameters that would be of value to substitute for one of your optical depth based parameters.

*Yes, we agree that the High IWC mask basically detects optically thick ice clouds, and the verification with DARDAR shows that optically thick ice clouds are predominantly the location where High IWC occurs.*
*Indeed, it should be kept in mind that the goal of the High IWC mask is to identify possible High IWC environments, preferably with a high probability of detection and a low false alarm rate.*

*As shown in the paper, the POD and FAR vary considerably depending on the particular combination of CPP parameters, even for similar CSI values (our chosen statistic to optimize). We have chosen one particular combination of CPP parameters, but other combinations could have been chosen. It is imaginable that with additional wishes at some point we might choose another combination of parameters.*

*It was, however, never the primary goal to minimize the number of CPP parameters. Furthermore, there is no harm in including CPP parameters that may not add much information.*

*Optimization regarding number of parameters, but also user requirements, is beyond the scope of this paper (see also next comment). And it should be kept in mind that with other or additional requirements, an approach very different from our pixel-based mask may or might have been chosen, like a morphological approach as done with RDT, or even a self-learning algorithm like a neural network.*

I appreciate the challenges you state regarding isolating HIWC in clouds and lack of understanding of how HIWC is generated. But this mask does not seem to be a product that the aviation community could use. No forecaster I know would alter flight routes based on this type of product.

*To what extent the High IWC mask is useful in forecasting and in the aviation community is beyond the scope of the paper and corresponding research questions as defined within the HAIC project.*

*We understand that one might be tempted to consider our paper in the light of the origins of the research topic (InService icing IS a potential aviation hazard).*

*However, we disagree on the notion that the product does not seem to be useful. Our personal experience with aviation industry is that it might very well be useful. So, we clearly differ in opinion.*

*If the scope of this paper had been to develop a data product for use in aviation, we surely would have included forecasters and the aviation community for feedback on user requirements. What either the authors or the referees think about its usefulness is in that context not very relevant.*

*The product seems more like a "ice cloud mask for moderate to high optical thickness cloud" than anything that could be useful for discriminating localized HIWC conditions. You're basically showing locations where HIWC would not be rather than where exactly it is. I have analyzed how your product would perform using every flight from the HAIC-HIWC project. I find that with your mask thresholds and use of TWC > 1 to define HIWC, your product would provide a 0.97 POD and 0.78 FAR In addition, if you're limited to solar zenith angle of < 60 then, you'd only have a product for a handful of hours a day, and only a few of these hours would actually have robust convection (i.e. intense storm activity typically begins no earlier than 2 PM). Given what I explain here and the thresholds you've selected as "optimal", please help me to understand how this product could be useful to the operational forecast community. Would you honestly expect aviators to fly around all the moderate optical thickness clouds you identify to "avoid HIWC"?*

*Whether or not the product is useful in an operational sense is beyond the scope of the paper, see previous comment.*

*Also, there is currently no aviation regulation regarding flying in High IWC/Ice Crystal conditions.*

*On the other hand, we see in analyses of actual flight tracks that other pixel-based CPP products are very useful for at least strategic flight planning, but possibly also for tactical flight planning when combined with short term predictions based on motion vectors.*

*Our analyses show that airliners and pilots make choices on their routes to avoid regions where CPP parameters exceed certain threshold values. As such, those results show how CPP products could be used: pilots would be advised to preferably avoid certain regions, similar to what is already currently being done using on board radar. The High IWC mask could be used in exactly the same way – if the recommendation would become to avoid possible High IWC environments. Note that there is no fundamental difference between how current on board radar is used compared to how a High IWC mask could be used. Also note that currently there are no formal restrictions regarding flying through thunderstorms (even though pilots are very strongly advised to avoid them).*

*https://www.faa.gov/documentlibrary/media/advisory_circular/ac%2000-24c.pdf*

*Finally, we would like to note that in the conclusions and recommendations with regard to the HAIC project we make the strong recommendation to provide the nowcasting community with more specific details about InService icing events. That detailed information turns out to be critical for developing more dedicated data products and services but is lacking. We believe that the nowcasting community is capable of providing such services, but that they (we) are hampered by the lack of specific detailed information.*

2) I still do not find the reasoning behind including RDT to be well explained nor are the findings worthwhile. Please explain the POD and FAR for RDT identification of HIWC. We know that based upon

your optical depth > 20 and cloud temp < 270 optimized HIWC mask thresholds, almost every convective anvil will be identified. Based upon the name of the product, RDT is designed to identify rapidly growing thunderstorms. This rapid development often occurs during the initial development phase where, as you indicate, the cells are small. So of course you should see some mismatch in the areal coverage of the product. You state that you are "comparing" the two products but the one scene you show isn't especially interesting aside from showing the areal coverage differences. In examples I personally have seen in talks and papers, RDT seems quite capable of detecting large anvils and embedded developing regions within them, not just newly developing small cells. I would think that, given how this product behaves and that it can extract cloud cooling and areal expansion, RDT may actually be better suited for HIWC detection than the mask you show here. Perhaps merging the two products (i.e. cloud temp < 270, optical depth < 20, within a cooling/expanding RDT object) would create a product more accurate and useful to forecasters. In summary, given the relatively weak analysis shown in the RDT section and tangential relationship to the primary focus of the paper (as it is currently written), I am not seeing how this section is relevant and request again that the RDT section be removed to improve the focus of the paper.

*The prime motivation for the HAIC and HIWC projects is that InService Icing events are frequently reported (far) outside of active convection, and are associated with the occurrence of High Ice Water Content. Once a satellite data product to detect High IWC becomes available, a relevant question is whether this product indeed identifies possible High IWC content regions (far) outside of active convective regions.*

*To this end, we use RDT, which has been specifically designed to identify active convection. Would the High IWC mask and RDT predominantly identify the same regions, then the justification for the development of the High IWC mask would be invalidated. It would also indicate that identification of High IWC as the primary focus of HAIC and HIWC would not be needed. The fact that there are significant differences between the High IWC mask and RDT suggests that the focus on High IWC rather than active convection is justified.*

*The fact that RDT uses a fundamentally different approach than CPP is an interesting additional point.*

*We have restructured the introduction and added some emphasis on the notion that prior to HAIC, there was no dedicated satellite data product for identifying potential InService Icing regions (which in the paper we explain is substituted with High IWC).*

*We also added the following to the section describing RDT (2.4):*

*"Because in service icing events are frequently associated with convective systems, it is valuable to evaluate the CPP performance against satellite data products specifically designed to identify active convection, and RDT is such a data product. A comparison with RDT will thus show whether or not the High IWC mask correlates well with the occurrence of active convection, or whether High IWC can also be expected to occur outside of active convective regions."*

Given what I describe above and below, I do not feel that what seems to be the core goal of the paper, namely the development, validation, and characterization of an HIWC mask was satisfactorily accomplished here. The authors raise many interesting points and are clearly excellent scientists. Yet the extraneous material throughout such as RDT and DARDAR-CPP comparisons, combined with double usage of optical depth in the mask and an end "optimized" mask product that basically classifies any moderately thick ice cloud as HIWC are major roadblocks for me that I don't think could be addressed in the limited time allotted for revision. Nevertheless, I will allow the authors one final opportunity to address the issues I raise. As I am already on the fence between major revision and reject, if the core issues I mention throughout are not addressed, I will reject the next round.

*We hope that – with the considerable modifications made to the paper and our answers here - we have clarified that the main goal of the paper was to identify High IWC environments, as exact conditions under which in service icing events occur are not well understood. The lack of understanding is well documented in the literature and actually forms the basis for the HAIC project (and USA sister project HIWC). High IWC serves as one condition needed for such events to occur (but clearly not the only one).*

*We also made an effort to highlight that InService icing is not primarily associated with deep/severe convection: if that were the case, it would never have been emerged as a severe aviation hazard. It would have been easy to avoid (on board radar). The particular lack of association of in service icing with deep/severe convection (where only 20% of reported incidents occurred) forms the whole fundament for the HAIC and HIWC projects. This lack of association could be related to the fact that airliners already generally avoid crossing deep convective cores, but that is difficult to prove.*

*We reiterate that to what extent our High IWC mask is useful for aviation is beyond the scope of the paper and beyond the scope of the corresponding HAIC work package. We clearly also differ in opinion with the referee about its usefulness: our experience is that it might be very useful in a practical sense. However, its usefulness is a question to be answered by users, not us or the referee.*

**Additional comments**

Line 37, replace IEC with IWC

*Done*

Line 58, recommend citing Smith et al. (JAMC, 2012) here

*Done. Also added a number of missing DOI's to the reference list.*

Lines 65-66, you note that 80% of in service events were related to ice crystals in anvils and 20% related to strong convection. Do you think that this is caused by the fact that aircraft are directed to avoid classic convection by air traffic control and via signals they see on their onboard radar. I'm guessing that if there were more convective core penetrations, your perception of where in service events are located would change. Please comment on this in the paper text.

*It appears quite likely/possible/plausible that the 80/20 ratio indeed is related to the ability of aircraft avoiding deep convection and severe weather: these can be picked up by the on board weather and thus be avoided. Whether or not that is the case is, however, difficult to prove.*

*We added the following comment.*

*"This may explain why only 20% of the reported events are associated with classic convection: such environments can generally be avoided by use of the on board weather radar."*

Line 80, please reference the partnership with North Americans via the HAIC-HIWC campaigns.

*Added the following comment: "field campaigns in cooperation with the American HIWC project"*

*Note that we know the FAA is a sponsor of HIWC, and that Boeing, NASA, and NCAR are involved, but that it is unclear to us whose project this actually is (there also does not appear to be a project website). Would be helpful for an appropriate reference.*

Line 85, if it is understood that 80% of events are outside of classical convection, then why was it a focus for HAIC to study core and near core regions?

*Added an explanation to the introduction that the goal of this paper – as well as the HAIC work package on satellite remote sensing – is to detect High IWC (> 1 g/m3), not deep convection, convective cores or convective overshoots.*

*Also keep in mind that the AIRBUS event database indicates that there are InService icing events that are not directly associated with convection and convective systems (see added comment in section 3.1).*

Line 93, it seems as though Grzych and Mason (2010) and Grzych et al and Brevin et al (SAE, 2015) have characterized in service icing pretty well. This contradicts your statement here that indicates that icing is not well understood or characterized. I recommend you revise your wording here.

*We agree that general atmospheric conditions associated with in service icing events have been identified in the aforementioned studies [see Fridlind et al., 2015; Ackermann et al., 2015]. However, detailed understanding of what actually happens with the aircraft and engines during icing events, and what the corresponding cloud (microphysical) conditions are, is lacking – which also provides the justification for the different HAIC/HIWC field campaigns and both the HAIC and HIWC projects. Due to this lack of understanding, this paper and the corresponding HAIC satellite remote sensing work package have been focusing on detection of IWC exceeding 1 g/m3, consistent with a consensus that high IWC is a necessary condition for the occurrence of in service icing [see Fridlind et al., 2015; Ackermann et al., 2015].*

*We made several changes to the introduction to address this issue and incorporate this level of understanding, but changes were made also in line with other remarks by the referee.*

Line 120, what are the "second and third HAIC flight campaigns?" I wouldn't expect many to know about this so I recommend you explain further or omit

*Omitted.*

Line 124, why is daytime only the focus here? Many flights occur at night especially over oceanic waters. Convection peaks in intensity at night over ocean and also over land associated with mesoscale convective systems that develop overnight. I understand that you want to utilize optical information to aid detection, and night would have IR only. Given what you say about the 60 degree solar zenith angle limitation, you're quite limited in the hours of day when your product can operate and intense convection is present. I request that you explain these additional limitations and prospects for night-time HIWC identification.

*We added a sentence explaining that in the HAIC project the choice was to use CPP, which critically depends on reflected solar radiation for retrieval of cloud properties and thus only provides daytime data.*

*Use of daylight means that more cloud physical properties can be retrieved (more accurately) than based on InfraRed channels only – in particular cloud optical thickness.*

*There are obviously possibilities to (a) retrieve cloud properties (at night) using only IR channels, and (b) construct a High IWC mask based on those measurements; or alternatively, (c) develop an object-related IR algorithm (like RDT) for detection of for example convection and design an associated High IWC mask. However, for both cases a separate study is needed, especially given the limitations of determining the condensed water path based on IR measurements only. Development of such products was also beyond the scope of the HAIC work package on satellite retrieval.*

*Hence, for now we stick with the HAIC objective of investigating whether daytime SEVIRI cloud parameters based on passive VIS-IR measurements can be used to identify High IWC environments.*

Lines 176-180, so are you saying here that the phenomenon of "small ice crystals" in storms commonly attributed to HIWC cannot be retrieved using satellite measurements? I see inclusion of Reff in your Mask Version 1, what is the purpose of including this field? I also recommend that these sentences be separated into another paragraph given that it is in an already long paragraph.

*Inclusion of Reff in Mask V1 – which was constructed based on an AIRBUS event database – was motivated by the hope that Reff might help in identifying possibly hazardous environmental conditions. This turned out not to be the case. For Mask V2, analysis of Reff revealed that it was not useful. This may not be that surprising, given that for Mask V2 we are looking at local cloud properties (maximum IWC > 1 g/m3). Although Reff provides some information on general cloud particle characteristics near the top of the cloud, it does not provide information on cloud particle characteristics at lower levels in the cloud.*

*We separated the paragraph and added a reference to the notion that Reff is in particular not representative for local cloud conditions.*

Line 184, please explain how these products differ from the "official" merged CALIPSO/CloudSat cloud microphysical products produced by the CloudSat Science Team (http://www.cloudsat.cira.colostate.edu/data-products/level-2c/2c-ice) and why you opted to use DARDAR rather than these? What are the advantages of DARDAR? With regards to DARDAR, I read in Austin et al (JGR, 2009) that the uncertainty of the CloudSat IWC retrieval can be up to 40%. Given this uncertainty, what you perceive to be an HIWC event may not be, and vice versa. This casts doubts on the data used as foundation for HIWC truth. I strongly request you address this in the paper text.

*We acknowledge that there are different CLOUDSAT/CALIPSO cloud profile products that could be used. However, DARDAR was chosen as it seemed appropriate and we could not identify reasons to prefer the official product over DARDAR. Furthermore, members from the DARDAR team were involved in HAIC as well, which means that communication regarding DARDAR and derived products was very simple. Finally, and for reasons of continuity, EarthCare, the European mission successor of CALIPSO/CLOUDSAT, will use the DARDAR algorithms. There is currently no corresponding American mission using official CLOUDSAT/CALIPSO algorithms.*

*Estimates of the error of DARDAR IWC are < 20%; [Delanoe et al., 2014]), which is better than the IWC from CloudSat alone. Only for very high DARDAR IWC (> 2 g/m3) substantial biases have been found that require care to be taken. However, for the use of DARDAR IWC in this study this bias is not relevant (the bias is such that DARDAR IWC > 2 g/m3 still means that the threshold value of 1 g/m3 will be exceeded)*

*Note that at the moment the vertical resolution of DARDAR (60 m) still is better than that of the "official" product (240 m).*

*A reference to the errors is added to section 2.4*

Also with your DARDAR discussions, you're mentioning height of maximum IWC quite a bit, i.e. line 438, Figure 9. If you look at Figures 1-5 of Setvak et al. (Atmos Res, 2013), you'll see that in and near deep convective core regions, the CloudSat signal attenuates in the layer nearest cloud top (their Fig 5 is the best example) which is thought to be due to the presence of graupel near cloud top in strong updrafts. So the apparent height of max IWC would be very high, but in actuality, this height may be further down into the cloud that cannot be seen by CloudSat. I strongly recommend you consider this in how you state your results and mention this observational characteristic in the paper text.

*We agree that in cases of very deep convective core regions there are attenuation issues, i.e. the real maximum IWC may be larger than the maximum IWC from DARDAR.*

*On the other hand, deep convective core regions will be regions with (very) high total condensed water for which it appears highly unlikely that the High IWC mask threshold of 1 g/m3 would not be exceeded.*

*For such cases, and where attenuation would play a role in properly characterizing the IWC profile, what will happen is that the actual maximum IWC will be higher than what is measured by DARDAR and further down in the cloud than what is measured by DARDAR.*

*So, for the verification (Figure 6/9) pixels would in reality have a higher IWCmax, and thus shift to the right in the distribution. However, since these are convective cores where IWCmax is already large, we are in the rightmost area of the figure where the rate of detection does not (strongly) depend on the IWCmax anymore. Such a shift in the distribution will have little impact on the performance of the mask.*

*In addition, deep convective cores (and especially overshoots) have limited spatial coverage, hence it appears unlikely that this has a large effect on the verification results (the active sensors CLOUDSAT/CALIPSO do not frequently encounter convective cores).*

*Overall, we do not expect that this effect has much impact on our results.*

Line 244, please provide a point of contact or a mechanism for one to acquire a list of these AIRBUS events for scientific research. I see an AIRBUS representative on the author list so I'd imagine that he could provide this information. The results from published journal articles are supposed to be reproducible and, without this list, one cannot verify the reproducibility of your results

*We added a remark to the Supplementary Information about the confidentiality of the results as well as an AIRBUS contact name (A. Calmels).*

*We – and AIRBUS - realize that in terms of reproducibility of results it is complicated to not have access to the database. However, these are conditions set by AIRBUS – it is their database, and there is no possibility to change that. Hence, the decision was made to not discuss the results of the analysis of the database events and the construction of the first HIWC mask in detail in the paper, but to briefly discuss the results in the SI.*

*The most important outcome of the analysis was that the database provided insufficient constraints on the CPP parameter threshold, and that was the primary motivation to adopt a different approach by using DARDAR data instead, which is what the paper is about.*

*Still, for completeness, we felt it relevant to mention the existence of the database and its role in the coming about of the paper.*

*The added remark about contacting AIRBUS for details about the database and providing the name of an AIRBUS employee who can be contacted should suffice for anyone wanting to know more details about the database.*

Line 255, you spend some time in the Supplemental section doing a pretty good job of presenting an interesting HIWC mask (I'd say a more interesting one than v2 mask), but then in lines 254-257, you almost immediately cast off both the training dataset and mask as being not useful. This seems very odd. . .why bother with describing the mask if you don't think the in service events are useful?

*As (hopefully) now better outlined in the paper, as well as discussed in some of the earlier comments and as well as in the previous comment, this paper focuses on detection of High IWC (IWC > 1 g/m3), and the mask is specifically designed to do so.*

*Originally we had hoped that the AIRBUS database of in-service events would contain sufficient information to characterize atmospheric conditions for which these events occur.However, we ended up with only nine (9) events, and for each event we only had information about its approximate location.This is obviously not sufficient for the generation of a HIWC mask. As explained above, we do think it is appropriate to mention the v1 mask in the supplement.*

Line 258, "was agreed upon", by whom? Please reword for clarity

*Added "was agreed upon by the HAIC project leaders".*

*Note that it was a decision discussed among and agreed upon by the entire project leaders group.*

Line 266, you say that a test dataset from 2008 was provided, but then only 31 orbits were analyzed? This is confusing; many more than 31 orbits were recorded in the entire year of 2008. Please explain in the paper text

*Changed the entire section to:*

*"A test dataset was constructed, for the year 2008 consisting of 31 daytime orbits in the year 2008, covering all months, randomly distributed throughout the SEVIRI disc, and containing a sufficient number of high IWC measurements within one orbit (see supplementary information table S1).*

Line 270, Figure 1, you use a combination of red and green to show where ice was observed in the profile. Red and green are an awful color combination for those that are color blind so I request that you consider this and recreate this and other figures that use red/green combos to highlight meaningful results.

*Changed to grey line with black-white dots indicating DARDAR profiles with ice in it*

Lines 270-328, I find that the material presented in these lines is extraneous and does not necessarily advance your goal of developing a HIWC mask. 278-288 talks about cloud top height assignment relative to DARDAR. Cloud top height was irrelevant to your optimized mask. Also, you spend a good bit of text comparing CPP to DARDAR IWC/IWP and effective radius considerations. Your goal in this paper was to develop a mask to discriminate HIWC using satellite observations. It really doesn't matter what CPP produces for CWP; all that matters is that you pick a threshold that gets the job of discriminating done well. As I'm reading the text prior to line 270, I'm engaged with the paper and its message, then line 270 comes along and I lose focus, waiting for the main point that never seems to come. Please explain the relevance for including this material and perhaps make more concise to stay on the primary message of the paper.

*This is the first time that a dedicated satellite-based High IWC product is presented. High Altitude Ice Crystals as a research topic is fairly new, and only emerged as a possible aviation hazard in the late 1990s and early 2000s.*

*Furthermore, research, results and understanding have grown organically within the HAIC project and the dedicated work packages on satellite measurements and detection of High IWC environments.*

*The SEVIRI CPP High IWC mask thus is a totally new product, and readers may not have general knowledge about for example DARDAR. We felt that introducing DARDAR without much more than a reference to other papers would be rather meager for those readers, so to speak.*

*Similarly, only during the last couple of years some intercomparison studies have appeared that compared measurements of cloud properties from passive (like SEVIRI) and active sensors (like DARDAR). Much of the work on identifying agreement and differences between those measurements is still work in progress, and often performed within the context of specific applications.*

*Because of these reasons, we thought it useful to spend some time on introducing DARDAR and presenting some general results about how DARDAR compares to SEVIRI-CPP. Like we similarly felt that it would be useful to also present the results on how the CPP High IWC mask v1 was constructed, which can be found in the SI.*

***[SUGGESTION]:*** *we would be willing to consider putting section 3.2 in the SI, rather than in the main body, and only briefly discuss results, like now is done for the SEVIRI-CPP High IWC mask v1. With the notion that we prefer to keep section 3.2 in the main text.*

Lines 348-353, I recommend rewording this paragraph as I'm not currently following what you're getting at here. You say "were then analyzed", what are you referring to? I don't see anything about steepness in the paper. Line 351 says "mask in the maximum IWC interval in the interval. . ." which reads a bit awkwardly.

*Changed "were then""to "will be*
*Changed "interval in the interval" to "*
*Changed line 351  to "the steepness of the increase in fraction of DARDAR profiles identified by the High IWC mask in the IWCMAX interval between 0.1 g/m3 and 1 g/m3"*

Lines 376-382, Figure 8a, I see that you have a ROC curve here, but your axes are backwards. FAR is usually along the Y-Axis so that the "area under the curve" can be analyzed to assess the quality of a

product. I recommend you reverse the axes here to make this plot consistent with how others in the validation community present their results.

*Changed figure 8a accordingly*

Line 401, as noted above, if one uses your thresholds to analyze product performance relative to 45 second averaged TWC > 1 from the IKP2 probe during HAIC-HIWC Darwin, you get a product with an estimated POD of 0.97 and FAR of 0.78. Obviously we're not using CPP results here but other cloud retrieval algorithms have been proven to provide comparable retrievals to CPP as evidenced in the Hamann GEWEX assessment paper. This is quite inconsistent with the results you get here which is a really big issue to contend with. How do you reconcile these differences between DARDAR based truth and IKP based truth?

*The problem of comparing in situ measurements (like IKP2 TWC) with passive remote sensing data like CPP is that the latter reflect vertically integrated cloud parameters (or of a part of the vertical cloud profile, depending on the optical depth of the cloud and whether UV/VIS or IR channels are used), rather than local cloud parameters.*

*We have made comparisons with IKP TWC for the Cayenne 2015 field campaign, which clearly shows that we can achieve a POD of better than 0.80, but an equally large FAR of about 0.80 (these results are indeed similar to those you report for HAIC-HIWC Darwin!).*

*The explanation for this behavior is that the possibility exists that local TWC (as measured by IKP) can be smaller than 1 g/m3 while elsewhere in the same cloud profile TWC can still exceed 1 g/m3. The CPP High IWC mask would identify such a cloud profile, but in the contingency statistics of in situ TWC this would result in a false detection.*

*Since we do not discuss field campaign data in this paper, we will not include a comment. However, results of the comparison between CPP and IKP for the HAIC-HIWC Cayenne 2015 field campaign will be reported in HAIC deliverable D32.4 (due date January 2017; unknown whether this deliverable will become public).*

Line 443-448. You mention that maximum IWC of 0.1 is "still quite high". My assessment of in-situ TWC data during HAIC-HIWC and column max IWC from RASTA retrievals is that 0.1 is about the 25th percentile of TWC observed during the HAIC/HIWC campaigns. My personal opinion is that 25th percentile is not particularly high.

*Based on the full 2008 DARDAR dataset (~25 million profiles), the maximum IWC in DARDAR profiles IWC > 1.0 refers to the $4^{th}$ percentile, IWC > 0.5 to the $7^{th}$ percentile, and IWC > 0.1 to the $20^{th}$ percentile, the latter consistent with what is reported above. Note that for much smaller subsamples of the 2008 DARDAR dataset these percentiles are very similar (including the DARDAR subsample used in this paper).*

*For IWC above 8 km altitude (cruising altitude) these percentiles are $0.5^{th}$, $1^{st}$ and $5^{th}$, respectively. Which is not surprising as the frequency of occurrence of high IWC decreases with altitude due to the decrease of temperature with altitude (which coincides with decreasing saturation vapor pressure of water)*

*Obviously what is meant with "quite" is subjective. Our thinking was that a maximum IWC of more than 0.1 g/m3 still represents a not too large minority of IWC conditions.*

*We added a remark that we consider a $20^{th}$ percentile being "quite high".*

Line 452, you may be seeing attenuation of CloudSat signal leading to higher IWC altitudes, which would indicate a more vigorous storm that could be easier to detect via satellite.

*Added a remark to line 452 mentioning the possibility of attenuation playing a role here.*

Line 470, I recommend a different term than "air masses" here

*Changed to "identify similar areas as potential high IWC environments"*

Line 563, I wouldn't exactly characterize the analysis here as "fine tuning" given the very liberal thresholds and double counting of optical depth used within the end product

*Changed to "shows that the mask is capable of detecting high IWC values > 1 g/m3 in the DARDAR profiles with a probability of detection of 60-80%."*

Line 577, after thunderstorms have ceased developing and they are decaying, wouldn't the risk of HIWC greatly decrease? So the fact that the HIWC mask identifies these may not be a good thing, correct?

*As discussed before, aviation industry reports that 80% of the reported in-service icing events is not directly associated with deep convective cores and/or active convection (meaning on board radar does not indicate such conditions).*

*The analysis of the AIRBUS event database suggests that such events can also occur in (cirrus) clouds which cannot directly be associated with a particular convective system.*

*The whole motivation for HAIC and its US sister project HIWC are the points discussed above: it has been frequently assumed that aviation problems with cloudy environments only or predominantly occur in or near deep severe convection, which is demonstrably incomplete. Hence the need for dedicated research.*

Line 603, where was the HAIC field campaign in 2016?

*At the time of writing it was still unclear where the 2016 field campaign would be held (essentially could be anywhere: Cayenne, Florida, Reunion, Indonesia, India, Australia were the possibilities at that time).*

*Changed to sentence to:*

*"The HAIC field campaigns in 2015 (Cayenne, French Guyana, South America) and 2016 (Darwin, northern Australia; Reunion, south-western Indian Ocean) …"*

**References.**

Ackerman, A. S., Fridlind, A. M., Grandin, A., Dezitter, F., Weber, M., Strapp, J. W., and Korolev, A. V.: High ice water content at low radar reflectivity near deep convection – Part 2: Evaluation of microphysical pathways in updraft parcel simulations, Atmos. Chem. Phys., 15, 11729-11751, doi:10.5194/acp-15-11729-2015, 2015.

Fridlind, A. M., Ackerman, A. S., Grandin, A., Dezitter, F., Weber, M., Strapp, J. W., Korolev, A. V., and Williams, C. R.: High ice water content at low radar reflectivity near deep convection – Part 1: Consistency of in situ and remote-sensing observations with stratiform rain column simulations, Atmos. Chem. Phys., 15, 11713-11728, doi:10.5194/acp-15-11713-2015, 2015.